# The polyol pathway is an evolutionarily conserved system for sensing glucose uptake

Hiroko Sano[1]*, Akira Nakamura[2], Mariko Yamane[3], Hitoshi Niwa[3], Takashi Nishimura[4], Kimi Araki[5,6], Kazumasa Takemoto[5,7], Kei-ichiro Ishiguro[7], Hiroki Aoki[8], Yuzuru Kato[9,10], Masayasu Kojima[1]

1 Department of Molecular Genetics, Institute of Life Science, Kurume University, Kurume, Fukuoka, Japan, 2 Department of Germline Development, Institute of Molecular Embryology and Genetics, and Graduate School of Pharmaceutical Sciences, Kumamoto University, Kumamoto, Kumamoto, Japan, 3 Department of Pluripotent Stem Cell Biology, Institute of Molecular Embryology and Genetics, and Graduate School of Medical Sciences, Kumamoto University, Kumamoto, Kumamoto, Japan, 4 Laboratory of Metabolic Regulation and Genetics, Institute for Molecular and Cellular Regulation, Gunma University, Maebashi, Gunma, Japan, 5 Institute of Resource Development and Analysis, Kumamoto University, Kumamoto, Kumamoto, Japan, 6 Center for Metabolic Regulation of Healthy Aging, Kumamoto University, Kumamoto, Kumamoto, Japan, 7 Department of Chromosome Biology, Institute of Molecular Embryology and Genetics, Kumamoto University, Kumamoto, Kumamoto, Japan, 8 Cardiovascular Research Institute, Kurume University, Kurume, Fukuoka, Japan, 9 Mammalian Development Laboratory, Department of Gene Function and Phenomics, National Institute of Genetics, Mishima, Shizuoka, Japan, 10 Department of Genetics, SOKENDAI, Mishima, Shizuoka, Japan

* sano_hiroko@kurume-u.ac.jp

**Data Availability Statement:** All relevant data are within the supporting information files, except for the RNA-seq original data files, which are uploaded to GEO (accession number: GSE195759).

## Abstract

Cells must adjust the expression levels of metabolic enzymes in response to fluctuating nutrient supply. For glucose, such metabolic remodeling is highly dependent on a master transcription factor ChREBP/MondoA. However, it remains elusive how glucose fluctuations are sensed by ChREBP/MondoA despite the stability of major glycolytic pathways. Here, we show that in both flies and mice, ChREBP/MondoA activation in response to glucose ingestion involves an evolutionarily conserved glucose-metabolizing pathway: the polyol pathway. The polyol pathway converts glucose to fructose via sorbitol. It has been believed that this pathway is almost silent, and its activation in hyperglycemic conditions has deleterious effects on human health. We show that the polyol pathway regulates the glucose-responsive nuclear translocation of Mondo, a *Drosophila* homologue of ChREBP/MondoA, which directs gene expression for organismal growth and metabolism. Likewise, inhibition of the polyol pathway in mice impairs ChREBP's nuclear localization and reduces glucose tolerance. We propose that the polyol pathway is an evolutionarily conserved sensing system for glucose uptake that allows metabolic remodeling.

## Introduction

The accurate sensing of ingested nutrients is vital for organismal survival. Animals need to sense quantitative and temporal changes in their nutritional status due to daily feeding to optimize metabolism. Glucose is the most commonly used energy source for animals and provides

**Funding:** This work was supported by the following grants: Japan Society for the Promotion of Science KAKENHI 17K07419 and 20K06647 (to HS), 16H06276 (AdAMS) (to HS and KA), 17H03686 and 21H02489 (to AN), 19H05743 and 20H03265 (to KI); The Program of the Joint Usage/Research Center for Developmental Medicine, IMEG, Kumamoto University (to HS); The Joint Research Program of the Institute for Molecular and Cellular Regulation, Gunma University (to HS); The Ichiro Kanehara Foundation for the Promotion of Medical Sciences and Medical Care (to HS); MEXT-Supported Program for the Inter-University Research Network for High Depth Omics, IMEG, Kumamoto University (to AN, MY, HN, and KI); MEXT-Supported Program for the Strategic Research Foundation at Private Universities (to HS and MK). The funders had no role in study design, data collection and analysis, decision to publish, or preparation of the manuscript.

**Competing interests:** The authors have declared that no competing interests exist.

**Abbreviations:** AEL, after egg laying; AR, aldose reductase; AUC, area under the curve; ChREBP, carbohydrate response element-binding protein; Dilps, *Drosophila* insulin-like peptide; InR, Insulin-like receptor; KEGG, Kyoto Encyclopedia of Genes and Genomes; Mlx, Max-like protein X; PPP, pentose phosphate pathway; Sodh, sorbitol dehydrogenase; TAG, triacylglyceride.

a good example of how they developed systems that achieve nutritional adaptation. Ingestion of glucose induces nutritional adaptation in the form of increases in glucose absorption and metabolism as well as lipogenesis to store excess nutrients, and inadequate adaptation might contribute to metabolic diseases such as obesity and type 2 diabetes. Glucose-dependent transcription is an important mechanism of glucose-induced nutritional adaptation [1], and such metabolic remodeling largely relies on a master transcription factor, carbohydrate response element-binding protein (ChREBP) [2]. ChREBP activates the expression of glycolytic and lipogenic genes with their obligated partner, Max-like protein X (Mlx), thereby storing excess nutrients in the form of lipids in the liver and adipose tissues [3]. MondoA, a paralog of ChREBP, functions in the skeletal muscle [4]. In *Drosophila*, the homologue of ChREBP/MondoA is encoded by a single gene, *Mondo*. Transcriptome analysis has shown that the Mondo-Mlx (also called Bigmax in *Drosophila*) complex induces global changes in metabolic gene expression in response to sugar uptake [5,6]. To achieve a metabolic shift, information on glucose availability must somehow be transmitted to ChREBP/MondoA.

Upon glucose uptake, ChREBP/MondoA is translocated to the nucleus, which is a pivotal step for ChREBP/MondoA activation. ChREBP/MondoA shuttles between cytoplasmic and nuclear compartments and is primarily localized at the cytoplasm in the basal state [4,7,8]. Glucose stimuli shift ChREBP/MondoA to the nuclei through the N-terminal domain containing a nuclear localization signal [9,10]. In parallel with this, derepression of transcriptional activation domain occurs and transcriptional activity is exerted. Although the precise mechanism is unknown, it has been thought that metabolites generated from glucose directly or indirectly regulate the nuclear localization of ChREBP/MondoA.

Using nuclear translocation and transcriptional activation as indicators, ChREBP/MondoA-activating sugars have been explored by administering candidate sugars to cultured cells. So far, several candidates such as xylulose 5-phosphate, glucose-6-phosphate, and fructose-2,6-bisphosphate have been identified [11–19]. These sugars are synthesized through either glycolysis or the pentose phosphate pathway (PPP) that branches off from glycolysis. Thus, cells were thought to detect blood glucose levels by assessing the activities of these two pathways. However, the levels of metabolites in these pathways remain mostly constant after glucose uptake partly due to the storage sugars [20]. Storage sugars such as glycogen are known to provide buffering action to prevent drastic changes in glucose metabolism; excessive nutrient uptake promotes the conversion of glucose-6-phosphate into glycogen, while starvation induces the breakdown of glycogen into glucose-6-phosphate. Moreover, glycolysis is tightly regulated by feedback control; glycolytic enzymes, including hexokinase working at the most upstream point in the pathway, are activated or inhibited by downstream metabolites [21]. Therefore, glycolysis and PPP are likely to be inadequate as real-time sensors to detect small changes in glucose concentration under normal physiological conditions. These findings suggest that the activation of ChREBP/MondoA involves a hitherto unrecognized pathway.

In this study, we show that the polyol pathway is involved in the activation of Mondo/ChREBP. The polyol pathway is a two-step metabolic pathway, in which glucose is reduced to sorbitol then converted to fructose [22]. Although physiological functions of this pathway remain elusive, the genes encoding polyol pathway enzymes are conserved from yeasts to humans, suggesting that it plays an important as yet unknown role across species. We demonstrate that the polyol pathway metabolites promote, and its mutations disturb nuclear translocation of *Drosophila* Mondo. The polyol pathway regulates Mondo/Mlx-target metabolic genes, leading to proper growth and physiology of *Drosophila* larvae. We further show that this pathway is required for glucose-responsive nuclear localization of ChREBP in hepatocytes and glucose tolerance in mice. Our results show that the polyol pathway is an evolutionarily conserved system for sensing glucose uptake that allows metabolic remodeling.

## Results

### Sorbitol feeding induces Mondo-mediated *CCHa2* expression

As a marker to assess what might activate Mondo/ChREBP, we chose a glucose-responsive hormone, CCHamide-2 (CCHa2). *CCHa2* has been suggested to be a target of Mondo, a *Drosophila* homologue of ChREBP/MondoA [6]. *CCHa2* is synthesized mainly in the fat body, an organ analogous to the mammalian liver and adipose tissues, in response to glucose ingestion [23]. The fat body is the prime organ of Mondo action as *Mondo* mutant phenotype can be largely rescued by restoring *Mondo* only in the fat body [5]. To examine whether Mondo activates *CCHa2* expression in the fat body, we knocked down *Mondo* specifically in the fat body. The knockdown reduced the expression of *CCHa2* under regular culture conditions (**Fig 1A**). It also reduced *CCHa2* expression upon glucose refeeding after 18-hour starvation, indicating that Mondo is required for acute induction of *CCHa2* expression in response to glucose ingestion (**Fig 1B**). This tissue-autonomous regulation by Mondo makes *CCHa2* an excellent marker for analyzing how sugars activate Mondo.

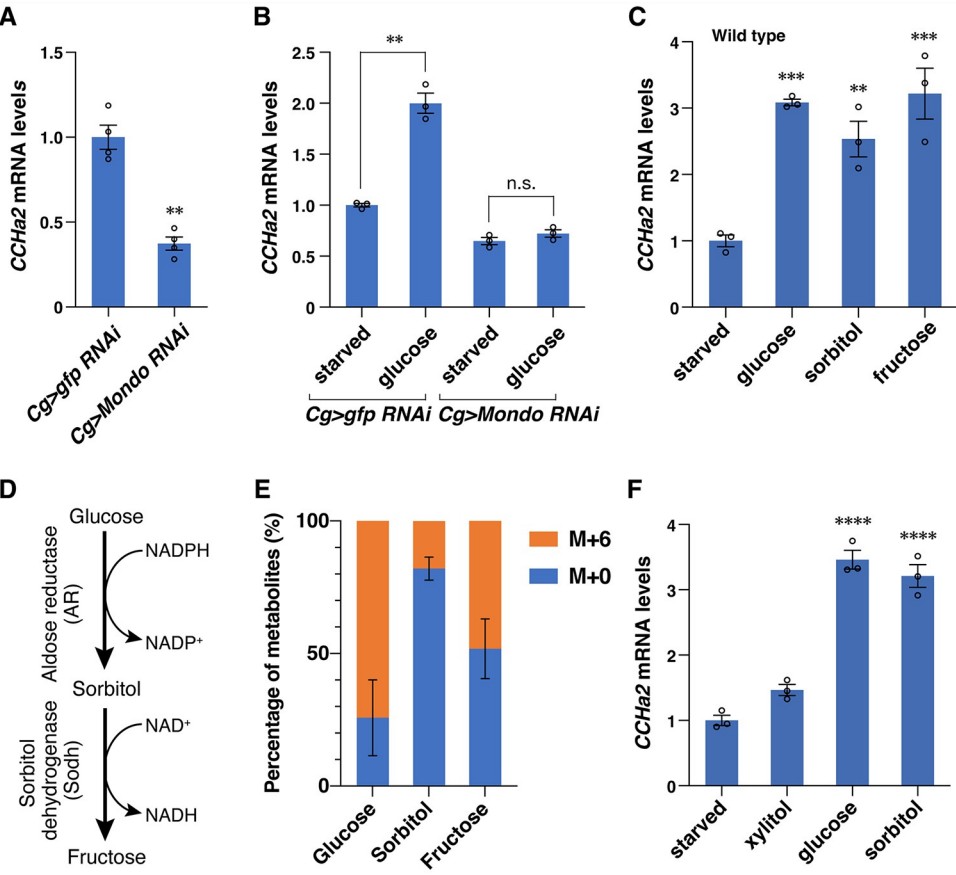

**Fig 1. Sorbitol feeding induces Mondo-mediated *CCHa2* expression.** (**A**) *CCHa2* mRNA levels in third instar larvae (72 hours AEL) in which *Mondo* was knocked down in the fat body using the *Cg-GAL4* driver. (**B**) *Mondo*-knockdown larvae were starved for 18 hours then refed with 10% glucose for 6 hours. *CCHa2* mRNA levels were quantified after refeeding. (**C**) Effects of different sugars on *CCHa2* expression. Starved wild-type larvae were refed for 6 hours with a 10% solution of the indicated sugars. (**D**) The polyol pathway. (**E**) The percentage of polyol pathway metabolites derived from $^{13}C_6$-labeled ingested glucose (M+6) and endogenous unlabeled glucose (M+0). (**F**) Starved wild-type larvae were refed for 6 hours with a 10% solution of the indicated sugars; 10 larvae per batch, $n = 3$ batches. Bar graphs show mean ± SE. n.s. $P > 0.05$; **$P < 0.01$; ***$P < 0.001$; ****$P < 0.0001$ (A–C and F); 3 larvae per batch, $n = 6$ batches. The bar graph shows mean ± SD (E). The data underlying the graphs can be found in **S1 Data**. AEL, after egg laying; AR, aldose reductase; Sodh, sorbitol dehydrogenase.

To identify metabolic pathways required for *CCHa2* expression, we examined the effects of sugars on *CCHa2* expression. We starved *Drosophila* larvae for 18 hours then refed them with several sugars. In addition to glucose and fructose [23,24], sorbitol was found to be capable of inducing *CCHa2* expression (**Fig 1C**). Because sorbitol is generated and metabolized exclusively by the polyol pathway [the Kyoto Encyclopedia of Genes and Genomes (KEGG) pathway database] [25–27], the induction of *CCHa2* expression likely involves metabolic reactions through the polyol pathway. In this pathway, glucose is converted to sorbitol by aldose reductase (AR, EC: 1.1.1.21), and then to fructose by sorbitol dehydrogenase (Sodh, EC: 1.1.1.14) (**Fig 1D**) [25–27]. It has been shown that the amount of sorbitol and fructose correlates with the amount of glucose in diets [28], and we demonstrate that sorbitol and fructose are synthesized from $^{13}$C-labeled ingested glucose (**Fig 1E**). These data indicate that the polyol pathway is functioning in *Drosophila*. While AR and Sodh are also predicted to transform xylose to xylulose via xylitol (the KEGG pathway database) [25–27], xylitol did not induce *CCHa2* expression significantly in wild-type larvae (**Fig 1F**). We thus reasoned that the role of the polyol pathway can be revealed by analyzing the requirement of AR and Sodh.

## The polyol pathway is required for proper larval growth and physiology

To create genetic tools to block the polyol pathway, we doubly mutated the putative *AR* genes, *CG6084* and *CG10638*, hereafter named *AR* mutants (**S1**and **S2** **Figs**). We also mutated sorbitol dehydrogenase (*Sodh*) genes, *Sodh-1* and *Sodh-2*, creating what we hereafter refer to as *Sodh* mutants (**S3 Fig**). Both *AR* and *Sodh* mutants showed metabolic phenotypes as predicted: Levels of sorbitol in the hemolymph were reduced in *AR* mutants and were increased in *Sodh* mutants (**S4 Fig**). Using the mutants, we examined whether the polyol pathway has any physiological function in *Drosophila* larvae. When mutant larvae were raised on a normal diet, *AR* and *Sodh* mutant larvae displayed a slight decrease in growth rate during larval stages, and significant pupation defects (**S5A–S5C Fig**) while the volume of mutant pupae and the body weight of mutant adults were indistinguishable from those of control animals (**S5D** and **S5E Fig**). Since *mlx* mutants have shown to be lethal on a protein-rich diet [5], we examined phenotypes of *AR* and *Sodh* mutants under the same culture condition. The mutant larvae showed a marked reduction in growth rate during larval stages, with only a small number of larvae becoming pupae (**Fig 2A** and **2B**). To examine whether the growth defects are due to downregulation of *Drosophila* insulin-like peptides (Dilps), we measured *dilp* mRNA levels and Dilp secretion in larvae at 96 hours after egg laying (AEL). *dilp5* mRNA levels were reduced in *AR* and *Sodh* mutants (**Fig 2C**). *dilp2* mRNA levels were increased (**Fig 2D**) and its secretion appeared normal as few Dilp2 signals remained in insulin-producing cells in mutants (**Fig 2E–2H**). We detected increased expression of *Insulin-like receptor (InR)* and *Thor* encoding *Drosophila* homologue of the initiation factor 4E-binding protein (4E-BP), which are indicators of insulin resistance [29], in *AR* mutants (**Fig 2I** and **2J**). Therefore, elevated *dilp2* expression is probably due to the development of insulin resistance, and insulin resistance may also contribute to growth defects in mutants. The polyol pathway mutants exhibited a decrease in whole larval triacylglyceride, and abnormal hemolymph glucose levels on a protein-rich diet (**Fig 2K–2P**). The phenotypic difference between *AR* and *Sodh* mutants is likely caused by AR's involvement in metabolic pathways other than the polyol pathway [25–27]. These metabolic phenotypes and growth defects were partially restored by the overexpression of *Mondo* in the fat body (**Fig 2K–2P**). It is unlikely that the rescue was incomplete due to the involvement of other tissues as the ubiquitous expression of Mondo did not significantly improve the rescue (**S6 Fig**).

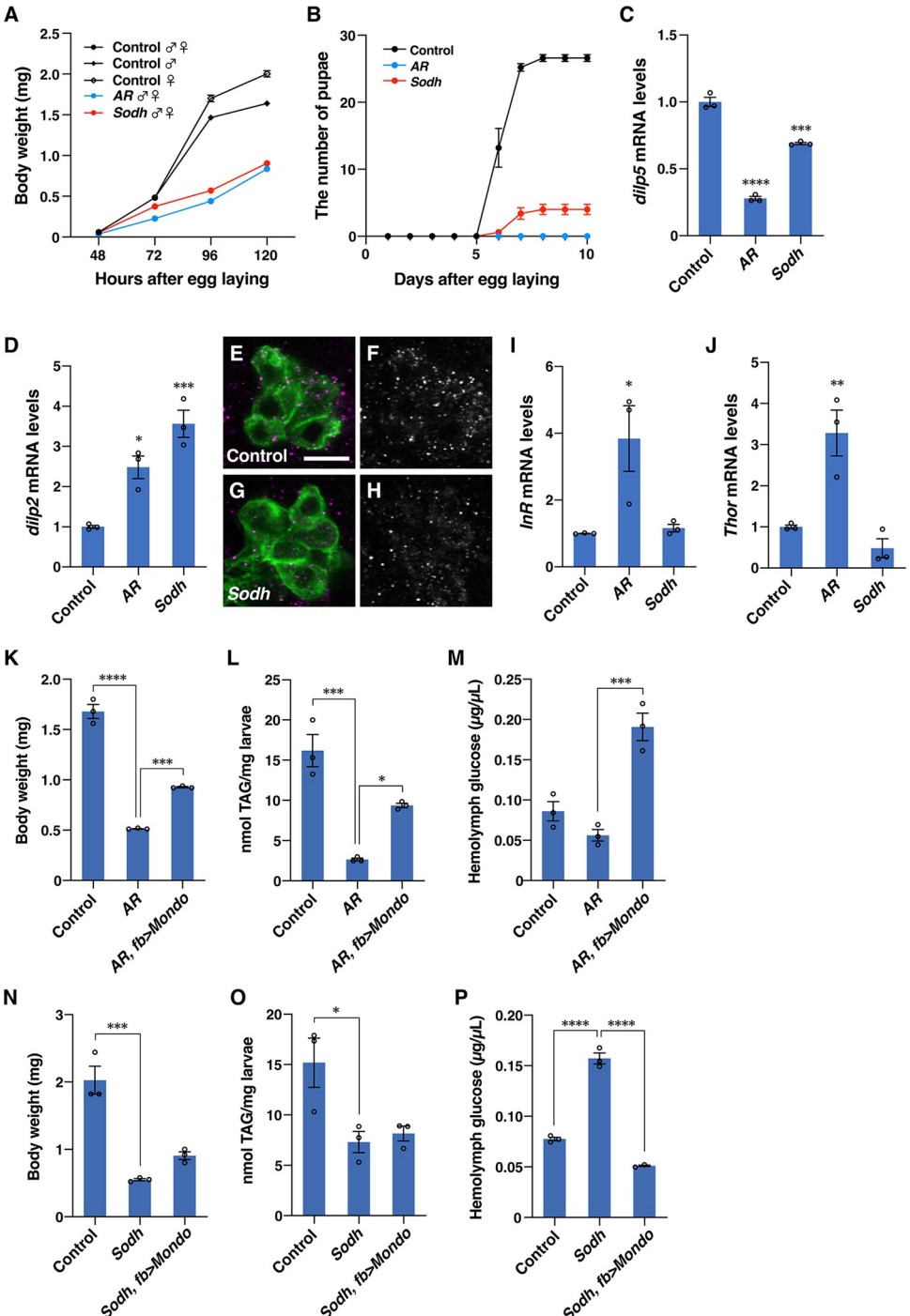

**Fig 2. The polyol pathway is required for proper larval growth and physiology.** (**A**) Larval body weight of *AR* and *Sodh* mutants reared on protein-rich diets; 10–50 animals per batch, *n* = 3 batches. The line graph shows mean ± SE. (**B**) Timing of pupal formation in *AR* and *Sodh* mutants. Thirty larvae were grown per vial and pupation was scored every 24 hours. *n* = 3 vials. The line graph shows mean ± SE. (**C–H**) Expression and secretion of Dilps in *AR* and *Sodh* mutant larvae. *dilp5* and *dilp2* mRNA levels were measured by RT-qPCR; 10 larvae per batch, *n* = 3 batches. Bar graphs show mean ± SE (C and D). Dilp2 was detected by anti-Dilp2 antibody (magenta), and insulin-producing cells were labeled by *dilp2>mCD8::GFP* (green) (E and G). Single-channel images of the Dilp2 signals are shown in F and H. Scale bar represents 10 μm. (**I, J**) *InR* and *Thor* mRNA levels were measured by RT-qPCR. (**K–P**) Body weight, whole body triacylglyceride, and hemolymph glucose levels in larvae of control, *AR* mutant, *AR* mutant with *Mondo* overexpression in the fat body (*AR, fb>Mondo*) (K–M), and control, *Sodh* mutant, *Sodh* mutant with *Mondo*

overexpression in the fat body (*Sodh, fb>Mondo*) (N–P); 5–10 larvae per batch, *n* = 3 batches. Bar graphs show mean ± SE. *$P < 0.05$; ***$P < 0.001$; ****$P < 0.0001$. The data underlying the graphs can be found in **S1 Data**. AR, aldose reductase; Dilps, *Drosophila* insulin-like peptide; RT-qPCR, reverse transcription quantitative PCR; Sodh, sorbitol dehydrogenase.

## The polyol pathway can mediate sugar-induced transcriptional alteration of Mondo/Mlx-target genes

The phenotypes of *AR* and *Sodh* mutants suggest that the polyol pathway regulates not only *CCHa2* but also a wide range of Mondo/Mlx-target genes. We thus tested whether the polyol pathway couples glucose ingestion to global transcriptional alteration through Mondo. Starved larvae were refed with either glucose or sorbitol, and expression levels of sugar-responsive Mondo/Mlx-target genes [6] (**S2 Data**) were quantified by RNA-seq analysis. Given that sorbitol is metabolized only through the polyol pathway, metabolites generated by the polyol pathway would be selectively increased in sorbitol-fed larvae, whereas metabolites of polyol, glycolytic, and PPP pathways would be increased in glucose-fed larvae. We detected a strong correlation between the changes induced by glucose and sorbitol (**Fig 3A**). Transcriptome changes upon sorbitol feeding were lost in the *Sodh* mutants (**Fig 3B**), confirming that sorbitol-induced gene regulation observed in wild type is dependent on the polyol pathway. Fructose, the end product of the polyol pathway, restored gene regulation in the *Sodh* mutants (compare **Fig 3C and 3D**), although the possibility that the influx of fructose into glycolysis participates in the rescue cannot formally be ruled out. These results show that the polyol pathway can regulate the expression of Mondo/Mlx-target genes.

We then examined whether the polyol pathway triggers Mondo/Mlx-mediated metabolic remodeling. We focused on genes encoding enzymes involved in glycolysis/gluconeogenesis, PPP, fatty acid biosynthesis, and glutamate and serine metabolism, many of which are under the control of Mondo/Mlx [6]. We observed similar expression patterns of these metabolic genes when wild-type larvae were fed with glucose or sorbitol (**Fig 3E**). Changes in metabolic gene expression were reduced in sorbitol-fed *Sodh* mutant larvae, but were restored in *Sodh* mutant larvae when fructose was fed. In particular, the levels of known Mondo/Mlx-target genes were remarkably restored (indicated in red or green in **Fig 3E**). These results show that the polyol pathway can induce the expression of various metabolic enzymes, leading to a metabolic remodeling in response to sugar ingestion. However, when starved *Sodh* mutant animals were fed with glucose, a considerable number of Mondo/Mlx-target genes including *CCHa2* were regulated properly (**Fig 3E–3G**). These results suggest that glucose-metabolizing pathways other than the polyol pathway can activate Mondo under starved conditions. In contrast, *CCHa2* expression was significantly reduced in *AR* and *Sodh* mutant larvae under normal feeding conditions with regular fly food containing 10% glucose (**Fig 3H**). These results suggest that the polyol pathway has differential requirements in different nutritional conditions; it is dispensable for Mondo/Mlx-mediated gene expression when glucose is provided after starvation but is required under normal nutritional conditions.

## The polyol pathway regulates nuclear localization of Mondo

The above results suggest that the polyol pathway is involved in a critical step in the activation of Mondo under normal physiological conditions. Therefore, we examined the effects of polyol pathway mutations on nuclear localization of Mondo. We tagged endogenous Mondo with the Venus fluorescent protein (**S7 Fig**), and observed intracellular localization of the Mondo:: Venus fusion protein under fed and starved conditions. We first examined Mondo::Venus localization in fat bodies dissected from normally fed third instar larvae. We utilized *ex vivo*

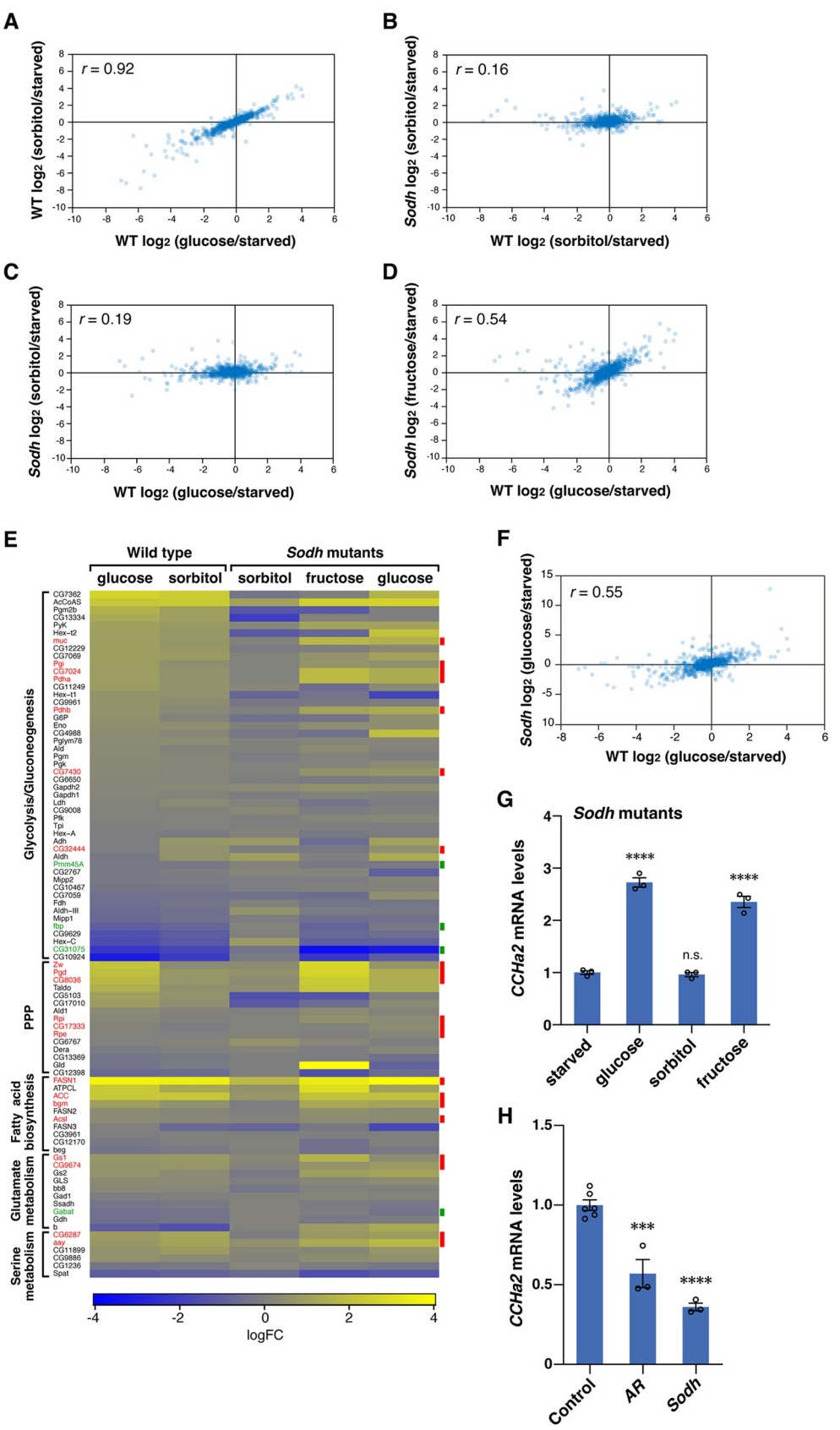

**Fig 3. The polyol pathway can mediate sugar-induced transcriptional alteration of Mondo/Mlx-target genes.** (A–D) Wild-type and *Sodh* mutant third instar larvae were starved for 18 hours, followed by refeeding for 6 hours with a 10% solution of indicated sugars. A comparison of expression changes of the Mondo/Mlx-target genes between glucose-fed and sorbitol-fed wild-type larvae (A), sorbitol-fed wild-type and *Sodh* mutant larvae (B), glucose-fed wild-type and sorbitol-fed *Sodh* mutant larvae (C), glucose-fed wild-type and fructose-fed *Sodh* mutant larvae (D). (**E**) The expression changes of metabolic genes. Genotype of larvae and fed sugars are indicated above. Known Mondo/Mlx-target genes are indicated in red (activated genes) or green (suppressed genes). (**F**) A comparison of expression changes of the Mondo/Mlx-target genes between glucose-fed wild-type and *Sodh* mutant larvae; 30 larvae per batch, *n* = 3 batches. Correlation coefficients (*r*) are indicated in the plots (A–D and F). (**G**) Effects of different sugars on *CCHa2* expression in *Sodh* mutant larvae. (**H**) *CCHa2* mRNA levels in the third instar larvae of polyol pathway mutants raised on a normal diet containing 10% glucose; 10 larvae per batch, *n* = 3 batches. Bar graphs show mean ± SE. n.s. $P > 0.05$; ***$P < 0.001$; ****$P < 0.0001$. The data underlying the graphs can be found in **S1 Data** and **S3–S8 Data**. AR, aldose reductase; Sodh, sorbitol dehydrogenase.

culture of fat bodies to minimize the effects of feeding conditions between larvae. It has been reported that mammalian ChREBP/MondoA displays nuclear localization when glucose concentrations are increased 5- to 10-fold [8,10–12,17,19,30,31]. Therefore, we compared the nuclear localization of the Mondo::Venus protein in fat bodies cultured in Schneider's *Drosophila* medium (hereafter referred to as the basic medium) that contains 11 mM glucose and those cultured in the same medium supplemented with 55 mM sugars. In the basic medium, 5.2% of Mondo::Venus signals were localized in the nuclei of wild-type fat body cells (**Fig 4A** and **4B**). When glucose, sorbitol, or fructose were added to the basic medium, the percentage of nuclear Mondo::Venus signals was increased to 14.6%, 12.7%, and 16.7%, respectively (**Fig 4A** and **4B**). In contrast, in the *AR* mutant or *Sodh* mutant fat body cells, glucose administration did not increase nuclear Mondo::Venus signals, suggesting that the polyol pathway required for the activation of Mondo (**Fig 4A**, **4C** and **4D**). Indeed, metabolites generated in the polyol pathway bypassed the requirements for *AR* or *Sodh* in the nuclear localization of Mondo (**Fig 4A**, **4C** and **4D**). These results indicate that the polyol pathway regulates the activity of Mondo by promoting its nuclear localization. To examine the localization of Mondo::Venus under starved conditions, larvae were starved for 18 hours then refed with glucose. The assay was performed at 1 hour after feeding, when the intestine was filled with sugar in most larvae. In contrast to the results in fat bodies from normally fed larvae, Mondo::Venus was translocated to the nucleus on the addition of glucose in *AR* and *Sodh* mutant fat body cells (**Fig 4E–4H**). These results are consistent with the conclusion from the gene expression analysis that the polyol pathway is dispensable for Mondo's function when glucose is provided after starvation but is required under normal nutritional conditions.

## The polyol pathway functions as a glucose-sensing system in mouse liver

To clarify whether the function of the polyol pathway in the sensing of glucose uptake is evolutionarily conserved, we knockout the *Sorbitol dehydrogenase (Sord)* gene, the only gene encoding Sorbitol dehydrogenase in mice (**S8A Fig**). *Sord* knockout mice showed normal growth and feeding (**Fig 5A** and **5B**). We investigated the nuclear localization of ChREBP in response to sugar ingestion in the liver of wild-type and *Sord* knockout mice. To remove ChREBP from the nuclei of the hepatocytes, we starved mice overnight. We orally administered sugar solution to starved mice and examined the intracellular localization of ChREBP in hepatocytes. We focused on pericentral hepatocytes (**S8B Fig**) as Sord is preferentially expressed [32,33]. Glucose or fructose ingestion promoted nuclear localization of ChREBP in wild-type mice (**Fig 5C–5E**, **5I–5K** and **5O**). In *Sord* knockout mice, glucose administration did not promote nuclear translocation of ChREBP (**Fig 5F**, **5G**, **5L**, **5M and 5O**), whereas fructose ingestion did (**Fig 5H**, **5N and 5O**). We then examined the metabolic phenotype of *Sord* knockout mice. They displayed similar phenotypes to those of *ChREBP* knockout mice: normal hepatic lipid

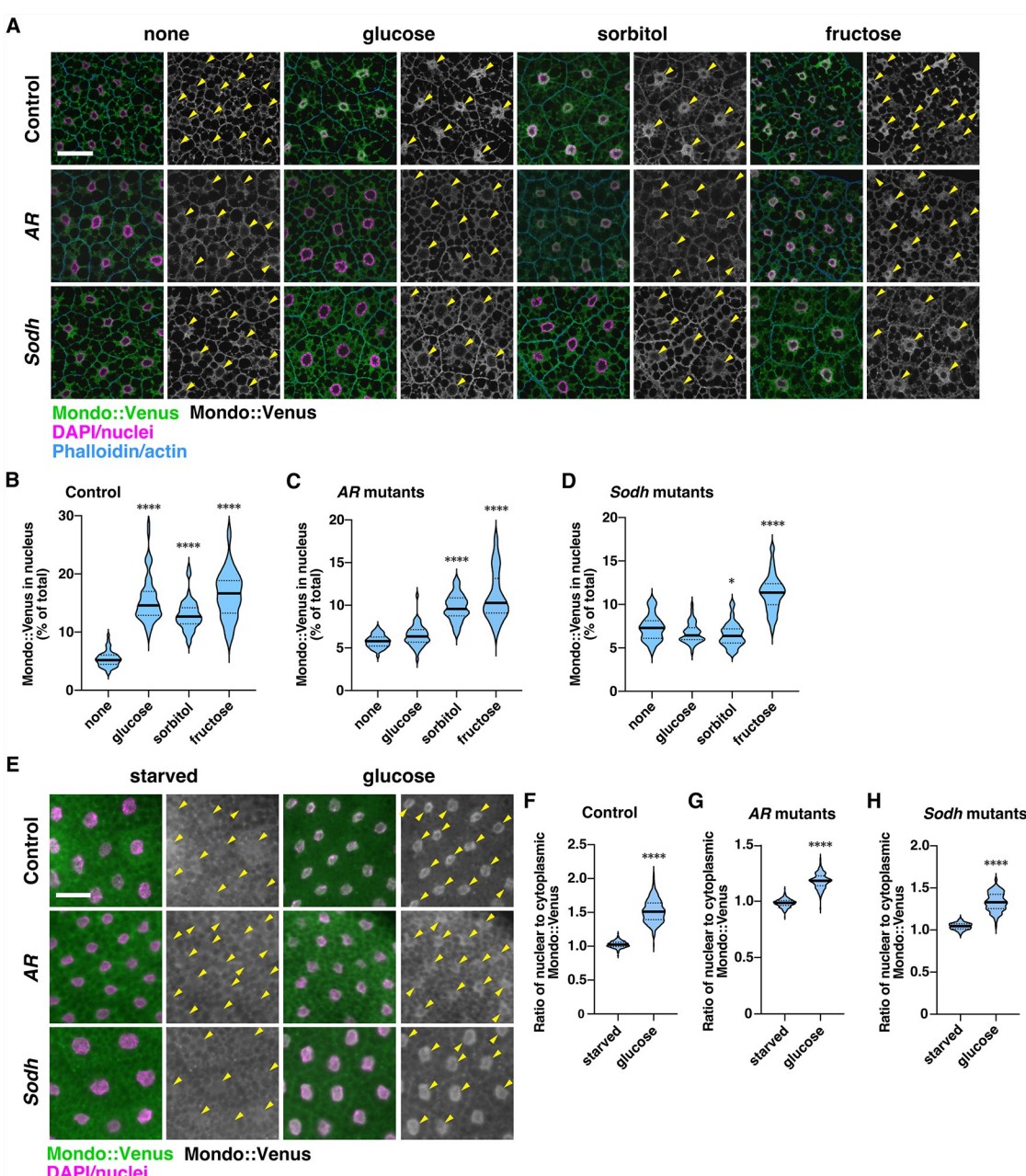

**Fig 4. The polyol pathway regulates nuclear localization of Mondo.** (**A**) Fat bodies dissected from normally fed Mondo::Venus knockin larvae were cultured in Schneider's *Drosophila* medium supplemented with 55 mM glucose, sorbitol, or fructose for 15 minutes. After culture, the fat bodies were fixed and stained with the following markers: anti-GFP antibody for Mondo::Venus (green), DAPI (magenta), and Rhodamine-conjugated phalloidin (blue). Single-channel images of the Venus signals are shown, with nuclei indicated by arrowheads. (**B–D**) The percentage of nuclear Mondo::Venus signal out of the total Mondo::Venus signal in a cell was quantified in the images. Solid and dotted lines in the graph show the median and quartiles, respectively. Images of 49 to 56 fat body cells per experiment were quantified. (**E**) Mondo::Venus larvae were starved for 18 hours then refed with 10% glucose for 1 hour. The localization of Mondo::Venus in the fat body cells was detected by Venus fluorescence (green), and nuclei were labeled with DAPI (magenta). Single-channel images of the Venus signals are shown, with nuclei indicated by arrowheads. (**F–H**) The ratio of average intensity of Mondo::Venus signals detected in the nucleus and cytoplasm in fat body cells. Solid and dotted lines in the graph show the median and quartiles, respectively. Images of 203 to 332 fat body cells per experiment were quantified. Scale bars in (A) and (E) represent 50 μm. $^*P < 0.05$; $^{****}P < 0.0001$. The data underlying the graphs can be found in **S1 Data**. AR, aldose reductase; Sodh, sorbitol dehydrogenase.

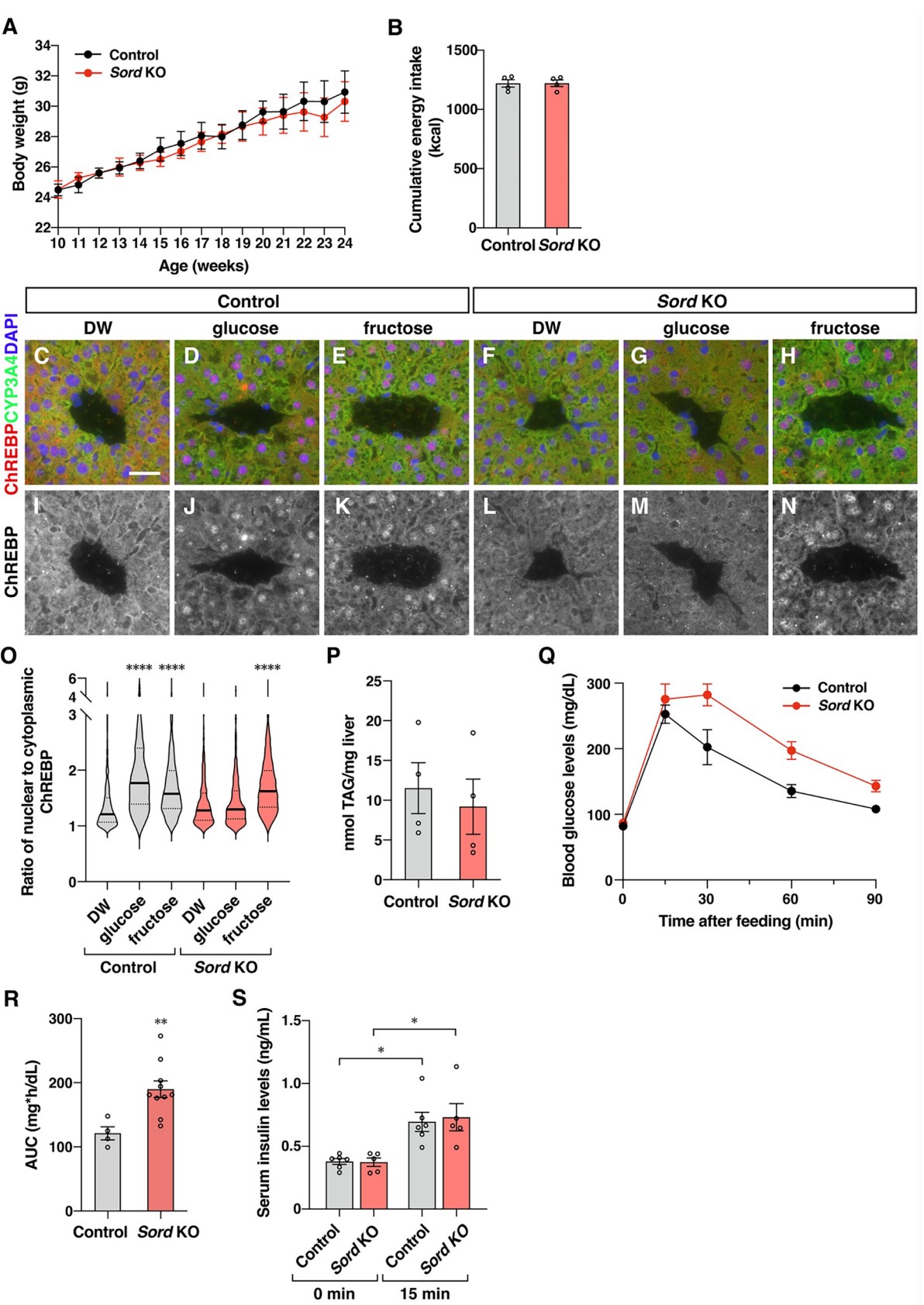

**Fig 5. The polyol pathway regulates nuclear localization of ChREBP in hepatocytes and glucose tolerance in mice.** (**A**, **B**) Body weight (A) and cumulative energy intake in control and *Sord* KO mice (B) were measured for 14 weeks. *n* = 4 for control, *n* = 4 for *Sord* KO. (**C–N**) Water (DW), glucose, or fructose were administered orally to starved control and *Sord* KO mice, and the localization of ChREBP in the hepatocytes was examined 15 minutes after administration. Liver slices were stained with the following markers: anti-ChREBP antibody (red), anti-CYP3A4 for pericentral hepatocytes (green), and DAPI for nuclei (blue). Scale bar represents 30 μm. (**O**) The ratio of average intensity of ChREBP signals detected in the nucleus and cytoplasm in the pericentral hepatocytes. Solid and dotted lines in the graph show the median and quartiles, respectively. Images of 539 to 910 hepatocytes per experiment were quantified. *n* = 3 animals per experiment. (**P**) The amount of hepatic triacylglyceride was quantified in control and *Sord* KO mice. (**Q**) Blood glucose levels were measured over a 90-minute period after glucose administration. (**R**) AUC was calculated relative to the fast blood glucose concentrations. *n* = 4 for control, *n* = 10 for *Sord* KO. (**S**) Plasma insulin levels were measured before and 15 minutes after glucose administration. *n* = 6 for control, *n* = 5 for *Sord* KO. All experiments were conducted using male mice. $^*P < 0.05$; $^{**}P < 0.01$; $^{****}P < 0.0001$. The data underlying the graphs can be found in **S1 Data**. AUC, area under the curve; ChREBP, carbohydrate response element-binding protein; KO, knockout.

accumulation (**Fig 5P**) and a delay in the recovery of blood glucose levels after oral glucose administration (**Fig 5Q** and **5R**). Serum insulin levels were comparable in wild-type and *Sord* knockout mice (**Fig 5S**). Therefore, observed glucose intolerance may be due to reduced insulin sensitivity, as in *ChREBP* knockout mice [3,34]. These results indicate that the polyol pathway has an important function in sensing glucose uptake in mouse liver, and its deficiency leads to impaired glucose tolerance. Thus, the polyol pathway is a common system for sensing glucose uptake in flies and mouse.

## Discussion

It has long been believed that the polyol pathway is almost silent, since the affinity of AR for glucose is very low compared to that of hexokinase for glucose. This pathway is thought to be activated only under hyperglycemic conditions, leading to diabetic complications [35]. However, genome research has revealed that genes encoding enzymes of the polyol pathway are conserved from yeasts to humans, suggesting that this pathway is important across species [25–27]. In this study, we revealed an evolutionarily conserved function of the polyol pathway in glucose sensing and organismal physiology.

### The significance of the polyol pathway in glucose sensing

To enable proper organismal adaptation to ingested glucose, the activity of the metabolic pathway(s) required for glucose sensing is expected to correlate with the levels of glucose in the body fluid. The polyol pathway appears to fulfill these conditions. First, the polyol pathway metabolizes glucose immediately after it enters the cell (**Fig 6A**) [25–27]. Second, the polyol pathway would be less affected by storage sugars. Glycogen, the major carbohydrate storage form in animal cells, is converted reversibly into glucose-6-phosphate according to nutrient status of the cell (**Fig 6A**). The adjustments to different nutritional states maintain constant glucose-6-phosphate levels, thereby aiding the stability of glycolysis and PPP regardless of the availability of glucose [20]. Third, no feedback control on the polyol pathway has been reported. This is a sharp contrast to glycolysis, in which several enzymes are subject to feedback control by downstream metabolites. Hexokinase acting at the most upstream point in glycolysis is tightly regulated by its product [21]. Phosphofructokinase 1, a rate-limiting enzyme of glycolysis, is also controlled by several downstream metabolites such as ATP, AMP, citrate, lactate, and fructose-2,6-bisphosphate [36]. These observations suggest that the polyol pathway could exhibit a linear response to glucose levels in the body fluid better than glycolysis and PPP under normal feeding conditions. Therefore, it is conceivable that the polyol pathway acts as a glucose-sensing system under conditions in which homeostasis of major glucose metabolic pathways is maintained by storage sugars and feedback control. Our results also show that the polyol pathway is dispensable for Mondo/Mlx-mediated gene expression and nuclear

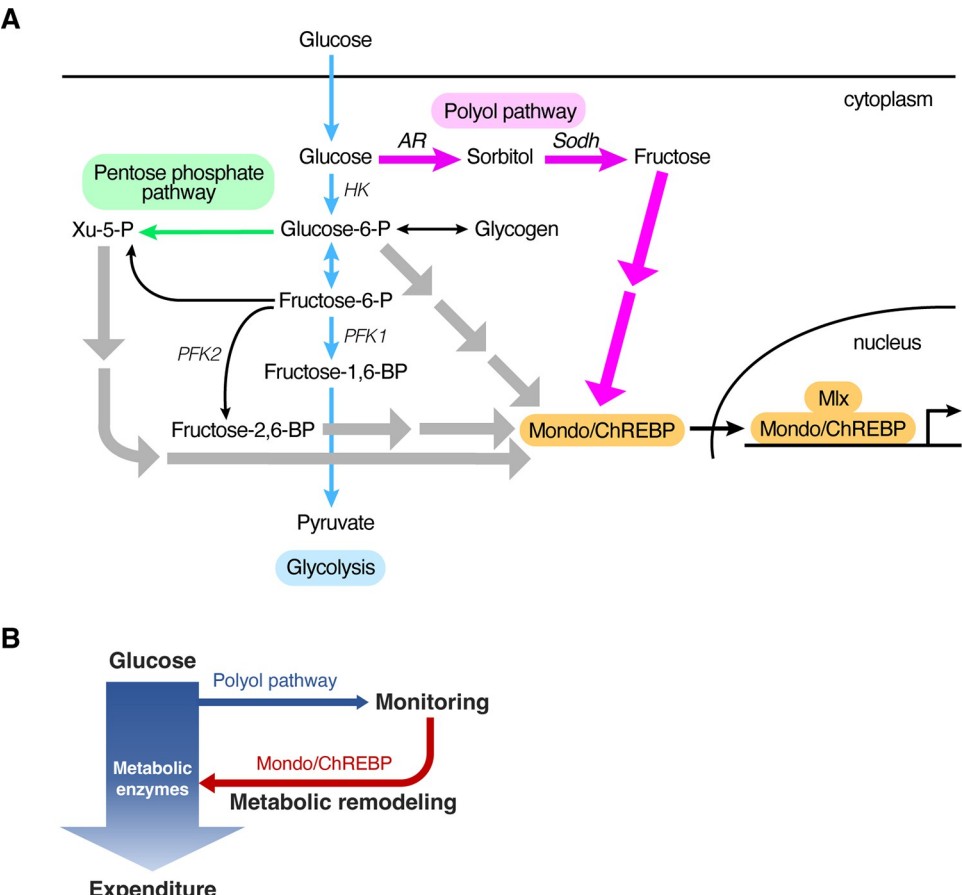

**Fig 6. The role of the polyol pathway.** (**A**) Metabolic pathways leading to Mondo/ChREBP activation. Glucose-6-phosphate (Glucose-6-P), xylulose-5-phosphate (Xu-5-P), and fructose 2,6-bisphosphate (Fructose-2,6-BP) were identified previously as MondoA/ChREBP-activating metabolites in mammalian cell culture systems (gray arrows) [2]. These metabolites are generated in glycolysis (blue arrows) or PPP (green arrow). Our study revealed that the polyol pathway (magenta arrows) has a significant contribution to activating Mondo/ChREBP in flies and mice (magenta arrows). (**B**) Physiological roles of the polyol pathway. The polyol pathway monitors glucose levels and regulates Mondo/ChREBP accordingly. This system allows animals to adjust metabolic activities to glucose availability and perform proper physiological functions. AR, aldose reductase; ChREBP, carbohydrate response element-binding protein; HK, hexokinase; Mlx, Max-like protein X; PFK1, phosphofructokinase 1; PFK2, phosphofructokinase 2; PPP, pentose phosphate pathway; Sodh, sorbitol dehydrogenase.

translocation of Mondo when the larvae were refed with glucose after 18-hour starvation (**Figs 3E–3G** and **4E–4H**), in which glycogen is completely consumed in the fat body [37]. In such situation, the activity of glycolysis and PPP could also reflect glucose uptake and function as a glucose sensor leading to Mondo activation. The existence of multiple glucose-sensing pathways might be related to the identification of glycolytic and PPP-derived metabolites as MondoA/ChREBP-activating sugars in mammalian cell culture systems (**Fig 6A**). Having various glucose-sensing systems would be beneficial for cells and organisms for their adaptation to different types of changes in nutritional conditions.

Our results suggest that fructose and fructose derivatives are good candidates for metabolite(s) that activates Mondo/ChREBP in sensing glucose uptake via the polyol pathway. It has been shown that the concentration of circulating fructose is acutely elevated upon glucose ingestion, probably due to the low basal concentration of fructose in the hemolymph [38]. Therefore, conversion of a portion of ingested glucose to fructose could be advantageous to

allow glucose detection, especially in hyperglycemic animals such as insects. Hyperglycemia is also observed in the mammalian liver to which dietary glucose is carried directly from the small intestine through the portal vein. We have shown that the polyol pathway is required for sensing glucose uptake in the mouse liver. The hepatic lobules are compartmentalized into regions with different metabolic functions along the porto-central axis: glycolysis and lipogenesis occur in the hepatocytes close to the central vein [39]. *Sord* mRNA is expressed with a peak in the pericentral hepatocytes [33], suggesting that the polyol pathway functions in the same region where glycolysis and lipogenesis occur and contributes to matching the activities of glycolysis and lipogenesis with glucose supply. On the other hand, whether fructose is released into the circulation and signals to other cells in the liver and other organs awaits further analysis.

### Insights into fructose-induced pathogenic mechanisms

The model that fructose or fructose derivatives activate Mondo/ChREBP explains the beneficial as well as harmful effects of fructose. We have shown that the polyol pathway, i.e., the presence of fructose, decreases the glycemic responses to oral glucose intake in mice (**Fig 5Q** and **5R**). Consistently, it has been shown that a small amount of fructose improves glucose in healthy and adults with diabetes [40–42]. On the other hand, it is well known that excessive fructose intake, as represented by high-fructose corn syrup, has adverse effects on human health [43–45]. It has been proposed that food-derived fructose is mostly cleared in the small intestine [46], and only a very small amount of fructose would be produced from glucose through the polyol pathway as glucose is a poor substrate for AR [35]. Therefore, a direct inflow of fructose to the liver caused by excessive fructose ingestion may mislead the cells into responding as if there has been very high amount of glucose ingestion, causing them to overactivate metabolic responses to glucose ingestion through ChREBP. Consistent with this, high-fructose ingestion in mice and rats is associated with increased ChREBP activity in the liver [47,48]. Our work lays the foundation for further important studies uncovering the molecular mechanisms linking abnormal sugar metabolism and disease development.

## Materials and methods

### Fry strains and dietary conditions

The following fly stocks were used: *Oregon-R (OR)*, *white (w)*, *y w*, *Cg-GAL4* (BDSC, RRID: BDSC_7011), *actin-Gal4* (BDSC, RRID: BDSC_4414), *UAS-Mondo RNAi* (VDRC, v109821), *UAS-Mondo*, *UAS-Mondo::HA* (FlyORF, F000486) [49], and *dilp2-GAL4>UAS-mCD8::GFP* [50]. $CG6084^{10-1}$, $CG10638^{10-1}$, $sodh1^{14-1}$, and $sodh1^{9-3}$ were generated using the CRISPR/Cas9 system (see below). Flies were raised at 25˚C on regular fly food containing (per liter) 40 g yeast extract, 50 g cornmeal, 30 g rice bran, 100 g glucose, and 6 g agar. Protein-rich food was prepared according to Havula and colleagues [5], with the following composition (per liter): 200 g yeast extract and 5 g agar.

### Mutagenesis and Venus knockin

The polyol mutants were generated using the CRISPR/Cas9 system as described in Gokcezade and colleagues [51]. The following sgRNA targets were used for the mutagenesis of the genes encoding AR and Sodh. Breakpoints of the mutants were determined as described previously [23,52] (**S1**–**S3** Figs). *CG6084* and *CG10638* were doubly mutated in *AR* mutants. *Sodh-1* and *Sodh-2* were doubly mutated in *Sodh* mutants.

*CG6084*: 5′-CCCCAAGGGTCAGGTCACCG

*CG10638*: 5′-GGCTACGAGATGCCAATTCT
*Sodh-1*: 5′-GATGTACACTACCTTGCACA
*Sodh-2*: 5′-GTGGGCAAGGTAGTGCACGT

The knockin of Venus at the C-terminus of the Mondo coding region was performed using the CRISPR/Cas9 system. The knockin vector was constructed by combining PCR-amplified left arm and right arm fragments for homologous recombination, and the Esp3I fragment of the pPVxRF3 vector (a gift from S. Kondo) containing Venus and 3xP3-dsRed-Express2 using NEBuilder HiFi DNA Assembly Master Mix (New England Biolabs, Ipswich, MA, USA). The combined fragment was cloned into pBluescript. The oligonucleotides used are as follows:

L-arm forward: 5′-GCTTGATATCGAATTCTGAACGACTGGAAATTTTGG
L-arm reverse: 5′-AGTTGGGGGCGTAGGGGGTGCATGCAGATTTGG
R-arm forward: 5′-TAGTATAGGAACTTCCGTTGATGCTGATGTCCTTG
R-arm reverse: 5′-CGGGCTGCAGGAATTCGAAAATGAGAGAAGATGGCGTA

The knockin vector was injected into *y w* embryo together with the sgRNA plasmid. The following sgRNA target was used.

5′-GGCCAGCATCCAAATCTGCA.

## RT-qPCR analysis

RT-qPCR was performed as described previously [23]. The following primers were used:

*CCHa2* forward: 5′-GCCTACGGTCATGTGTGCTAC
*CCHa2* reverse: 5′-ATCATGGGCAGTAGGCCATT
*dilp2* forward: 5′-GTATGGTGTGCGAGGAGTAT
*dilp2* reverse: 5′-TGAGTACACCCCCAAGATAG
*dilp5* forward: 5′-TGTTCGCCAAACGAGGCACCTTGG
*dilp5* reverse: 5′-CACGATTTGCGGCAACAGGAGTCG
*InR* forward: 5′-TGAGCATGTGGAGCACATCAAGATG
*InR* reverse: 5′-CGTAGGAGATTTTCTCGTTTGGCTG
*Thor* forward: 5′-TGATCACCAGGAAGGTTGTCATCTC
*Thor* reverse: 5′-GAGCCACGGAGATTCTTCATGAAAG
*rp49* forward: 5′-AGTATCTGATGCCCAACATCG
*rp49* reverse: 5′-CAATCTCCTTGCGCTTCTTG

## RNA-sequencing

Third instar larvae (72 hours AEL) were starved for 18 hours on water agar plates. The larvae were refed on agar plates containing 10% of indicated sugar for 6 hours. Sugar plates were supplemented with 1% Brilliant Blue to visualize larval sugar ingestion. Total RNA from whole larvae was extracted using the PureLink RNA Mini Kit (Thermo Fisher Scientific, Waltham, MA, USA). The library was constructed using the TruSeq Stranded mRNA LT Sample Prep Kit (Illumina, San Diego, CA, USA). RNA-seq was performed with NextSeq 500 (Illumina), targeting at least 14 million, single-end reads of 75 bp in size. The quality of the reads was assessed using FastQC (version 0.11.5). The reads were mapped to the FlyBase reference genome (Dmel Release 6.19) using Tophat2 [53]. Transcript abundance and splice variant identification were determined using Cufflinks [54], and differential expression analysis was performed using CuffDiff [54]. We confirmed that gene expression patterns correlate well in starved wild-type and *Sodh* mutant larvae, indicating that the observed differences in sugar-dependent gene expression between wild type and mutants are not due to variations in their genetic background (**S9 Fig**).

## Analysis of animal weight, volume, and developmental timing

Analysis of animal weight and developmental timing were performed as described previously [23]. The volume of pupae was measured using the formula $4/3\pi(L/2)(l/2)^2$ ($L$ is length and $l$ is diameter).

## Measurement of triacylglyceride and glucose

For measurement of triacylglyceride (TAG) concentration in the whole body, third instar larvae (96 hours AEL) were homogenized in water with 5% NP-40. The homogenate was heated at 90°C for 5 minutes and then mixed by vortex, which was repeated twice. The homogenate was centrifuged at 16,000 $g$ for 2 minutes, and the supernatant was used for TAG quantification using the Triglyceride Quantification Colorimetric/Fluorometric Kit (BioVision, Milpitas, CA, USA). For measurement of hemolymph glucose levels, third instar larvae (96 hours AEL) were rinsed with water and dried on filter paper. The cuticle was torn by forceps to release the hemolymph on a Parafilm membrane. The 1 µl of hemolymph was diluted with 9 µL of Tris buffered saline (pH 6.6) and immediately heated at 70°C for 5 minutes. The hemolymph solution was centrifuged at 16,000 $g$ for 1 minute, and the supernatant was used for glucose quantification using the Glucose Colorimetric/Fluorometric Assay Kit (BioVision).

## Tracing experiments

Third instar larvae (96 hours AEL) were transferred to a diet containing 5% U-$^{13}C_6$-D-Glucose (Cambridge Isotope Laboratories, Tewksbury, MA, USA), 1% Brilliant Blue, and 2% agar. Brilliant Blue was used to visually monitor food ingestion. Larvae with uniform amount of Brilliant Blue in their midgut were selected for metabolomics analysis to prevent any biases caused by differences in feeding. Frozen samples in 1.5 ml plastic tubes were homogenized in 300 µl cold methanol with 1× ϕ3 mm zirconia beads using a freeze crusher (TAITEC, Koshigaya, Saitama, Japan) at 41.6 Hz for 2 minutes. The homogenates were mixed with 200 µl methanol, 200 µl $H_2O$, and 200 µl $CHCl_3$ and vortexed for 20 minutes at room temperature. The samples were centrifuged at 15,000 rpm (20,000 $g$) for 15 minutes at 4°C. The supernatant was mixed with 350 µl $H_2O$ and vortexed for 10 minutes at room temperature. The samples were centrifuged at 15,000 rpm for 15 minutes at 4°C, and the aqueous phase was collected. The samples were dried down in a vacuum concentrator, re-dissolved in 2 mM ammonium bicarbonate (pH 8.0), and analyzed by LC–MS/MS.

Chromatographic separation and mass spectrometric analyses were performed essentially as described previously [55]. Chromatographic separation was performed on an ACQUITY BEH Amide column (100 mm × 2.1 mm, 1.7 µm particles; Waters, Milford, MA, USA) in combination with a VanGuard precolumn (5 mm × 2.1 mm, 1.7 µm particles) using an Acquity UPLC H-Class System (Waters). Elution was performed at 30°C under isocratic conditions (0.3 ml/minute, 70% acetonitrile, and 30% 10 mM ammonium bicarbonate (pH 10.0)). The mass spectrometric analysis was performed using a Xevo TQD triple quadrupole mass spectrometer (Waters) coupled with an electrospray ionization source in the negative ion mode. The multiple reaction monitoring transitions were as follows: $m/z$ 179.1 to 89.0 for $^{12}C_6$-fructose and -glucose, $m/z$ 185.1 to 92.0 for $^{13}C_6$-fructose and -glucose, $m/z$ 181.1 to 89.0 for $^{12}C_6$-sorbitol, and $m/z$ 187.1 to 92.0 for $^{13}C_6$-sorbitol.

## Metabolic assays using gas chromatography–mass spectrometry

Third instar larvae (96 hours AEL) were rinsed with water and dried on filter paper. The cuticle was torn by forceps to release the hemolymph on a Parafilm membrane. The 1 µl of hemolymph was collected and immediately quenched by mixing with 300 µl of cold methanol. The

samples were further mixed with 200 μl of methanol, 200 μl of $H_2O$, and 200 μl of $CHCl_3$ and vortexed for 20 minutes at room temperature. The samples were centrifuged at 20,000 *g* for 15 minutes at 4˚C. The supernatant was mixed with 350 μl of $H_2O$ and vortexed for 10 minutes at room temperature. The samples were centrifuged at 20,000 *g* for 15 minutes at 4˚C. The aqueous phase was collected and dried in a vacuum concentrator. Methoxyamine pyridine solution [20 mg/ml methoxyamine hydrochloride (FUJIFILM Wako, Osaka, Japan) in pyridine] was added to the dried residue to re-dissolve and oximated for 90 minutes at 30˚C. Then, MSTFA + 1%TMCS (Thermo Fisher Scientific) was added and incubated for 60 minutes at 37˚C for trimethylsilylation. The derivatized metabolites were analyzed by an Agilent 7890B GC coupled to a 5977A Mass Selective Detector (Agilent Technologies, Santa Clara, CA, USA) under the following conditions: carrier gas, helium; flow rate, 0.8 ml/minute; column, DB-5MS + DG (30 m × 0.25 mm, 0.25 μm film thickness, Agilent Technologies); injection mode, 1:10 split; inlet temperature, 250˚C; ion source temperature, 230˚C; quadrupole temperature, 150˚C. The column temperature was held at 60˚C for 1 minute, and then increased to 325˚C at a rate of 10˚C/minute. The detector was operated in the electron impact ionization mode. The Agilent-Fiehn GC/MS Metabolomics RTL Library was used for metabolite identification [55]. Metabolites were detected in SIM mode and the peak area of interests was analyzed by the QuantAnalysis software (Agilent Technologies).

## Detection of sugar-dependent nuclear translocation of Mondo::Venus in fat body cells

Fat bodies were dissected and cultured in Schneider's *Drosophila* medium that contains 11 mM glucose. We compared the nuclear localization of Mondo::Venus in fat bodies cultured in Schneider's *Drosophila* medium and in fat bodies cultured in the same medium supplemented with 55 mM glucose, sorbitol, or fructose. To detect the nuclear localization of Mondo in starved conditions, larvae were starved for 18 hours. The larvae were refed on agar plates containing 10% glucose for 1 hour. Sugar plates were supplemented with 1% Brilliant Blue to visualize larval glucose ingestion. Mondo::Venus signals in the fat body were visualized as follows.

## Immunofluorescence and image analysis of fat bodies

To detect Mondo::Venus in cultured fat body cells, fat bodies were fixed with 4% paraformaldehyde in PBS for 30 minutes and stained with rabbit anti-GFP polyclonal antibody (Thermo Fisher Scientific, 1:1,000) and Alexa Fluor 488-conjugated anti-rabbit-IgG (Thermo Fisher Scientific, 1:500). Nuclei and cortical actin were labeled with DAPI (Thermo Fisher Scientific, 1 μg/mL) and Rhodamine-conjugated Phalloidin (Thermo Fisher Scientific, 1:100), respectively. After staining, fat bodies were mounted in VECTASHIELD Mounting Medium (Vector Laboratories, Newark, CA, USA) and imaged with TCS SP8 confocal microscope using a Plan-Apochromat 63× oil-immersion objective lens (Leica Microsystems, Wetzlar, Germany) or Fluoview FV1000 confocal microscope using a UPlanSApo 60× water-immersion objective lens (Olympus, Tokyo, Japan). Images of the fat body were analyzed using the ImageJ2 software (version 2.0.0-rc-43, NIH). Cell and nucleus contours were traced with the freehand tool on the image. Signal intensities within each region were measured, and then the percentage of nuclear Mondo was calculated. To detect glucose-dependent Mondo::Venus localization in starved fat body cells, fat bodies were fixed with 4% paraformaldehyde in PBS for 15 minutes. Nuclei were labeled with DAPI (Thermo Fisher Scientific, 1 μg/mL). Venus fluorescence and DAPI signals were detected with Biorevo BZ-9000 fluorescent microscope using a Plan-Apochromat 40× lens (Keyence, Osaka, Japan). Nucleus contour was traced using a custom program written for the R software environment (www.r-project.org) with the EBImage package

[56]. The custom program is available from the following website (https://github.com/yukatonig/fatbody). To detect Dilp2 in insulin-producing cells, larval brains containing the *dilp2>mCD8*::*GFP* reporter were stained with rabbit anti-Dilp2 and mouse anti-GFP antibodies as described previously [23].

## Generation of *Sord* knockout mouse

*Sord* knockout mouse was generated as described previously by introducing the Cas9 protein, tracrRNA, crRNA, and ssODN into C57BL/6N fertilized eggs [57]. For generating the *Sord Δex3-9* allele, the synthetic crRNA was designed to direct GAGACAAAGGAAACACGTGA (GGG) in the intron 2 and AATCACAGTAGAACACACAA(AGG) in the exon 9. ssODN: 5′-TTCTTCATAAGTCAGCCCCACTCTCTGGCAATCACAGTAGTTTATTTATTTATGA GGGAAAGGCGAACCTTCCATTGCTCTCAGAAGTGCTA was used as a template for homologous recombination. The genome of targeted F0 mice was amplified by PCR using the Sord 13357- and Sord -30158 primers. A 1,052-bp fragment was amplified from the genome of the *Sord Δex3-9* allele. The PCR amplicons were sequenced using the Sord 13815- primer. F0 mice were backcrossed with C57BL/6N to establish the *Sord Δex3-9* line.

 Sord 13357-: 5′-GCAGTCTCTGGCCAGTTTTC
 Sord -30158: 5′-TTGCCTGTGAGTGACTCTGG
 Sord 13815-: 5′-CGGTTTCCTTTGGAATCTCA

 *Sord* KO mice were maintained under a temperature- and humidity-controlled specific pathogen-free conditions (22 ± 2°C, 50 ± 20%) on a 12-hour dark/12-hour light cycle and were allowed ad libitum access to normal laboratory chow and water. Body weight and food intake were measured every week. The animal protocol was approved by the Kurume University Institutional Animal Care and Use Committee (approval number: 2018–242, 2019–075, 2020–158, and 2021–209).

## Oral administration of sugar solution in mice

Eight-week-old male mice were starved for 16 hours before the experiment. Mice were weighed and given a sugar solution (2 g/kg body weight glucose or 3 g/kg body weight fructose) using a plastic feeding needle ϕ1.18 × 38 mm (AS ONE, Osaka, Japan). Blood glucose levels were measured before and after sugar administration. The liver was removed 15 minutes after sugar administration, embedded in the Tissue-Tek O.C.T. compound (Sakura Finetek, Osaka, Japan) and frozen in liquid nitrogen-cooled isopentane.

## Immunofluorescence and image analysis of mouse hepatocytes

Immunofluorescence staining was performed on 5-μm frozen section of the liver. The frozen sections were fixed with 4% paraformaldehyde in PBS for 10 minutes at 4°C. ChREBP was detected with rabbit anti-ChREBP polyclonal antibody 1:100 (Novus Biologicals, Centennial, CO, USA) and Cy3-conjugated anti-rabbit-IgG 1:500 (Jackson ImmunoResearch, Bar Harbor, ME, USA). Pericentral hepatocytes were labeled with mouse-anti-CYP3A4 antibody 1:300 (Proteintech Group, Rosemont, IL, USA) and Alexa Fluor 488-conjugated anti-mouse-IgG 1:500 (Thermo Fisher Scientific). Nuclei were labeled with DAPI (Thermo Fisher Scientific, 1 μg/mL). After staining, the sections were mounted in VECTASHIELD HardSet Antifade Mounting Medium (Vector Laboratories) and imaged with Biorevo BZ-9000 fluorescence microscope using a Plan-Apochromat 40× objective lens (Keyence). Image analysis was performed using ArrayScan XTI (Thermo Fisher Scientific) and the FlowJo 10 software (Becton Dickinson, Franklin Lakes, NJ, USA). ChREBP signals in CYP3A4-positive pericentral hepatocytes were quantified.

## Measurement of hepatic triacylglyceride

Hepatic triacylglyceride levels were determined using the Triglyceride Quantification Colorimetric/Fluorometric Kit (BioVision).

## Glucose tolerance test

Eight-week-old male mice were starved for 16 hours before the experiment. Mice were weighed and given glucose solution (2 g/kg body weight) using a plastic feeding needle $\phi 1.18 \times 38$ mm (AS ONE). Blood glucose levels were measured over a 90-minute period after glucose administration using a blood glucose monitoring meter (SANWA KAGAKU KENKYUSHO, Nagoya, Aichi, Japan). Area under the curve (AUC) was calculated relative to the fasted blood glucose concentration. Plasma insulin levels were measured before and 15 minutes after glucose administration using the Morinaga Ultra Sensitive Mouse Insulin ELISA Kit (Morinaga, Tokyo, Japan).

## Statistics

Two-tailed $t$ test was used to evaluate the significance of the results between 2 samples. For multiple comparisons, Tukey–Kramer or Dunnett test was used. A $p$-value of less than 0.05 was considered statistically significant.

## Supporting information

**S1 Fig. Generation of the *CG6084* mutant allele.** (**A**) CRISPR-mediated mutagenesis of the *CG6084* gene. A sgRNA was designed for the sequence within the exon common to the *CG6084* isoforms. The genomic map was adapted from FlyBase (http://flybase.org). (**B**) Breakpoint of the *CG6084*[10-1] allele. The *CG6084*[10-1] mutation caused a frameshift (yellow) leading to a premature termination in all isoforms of the CG6084 protein. The mutant proteins lack most of the catalytic domain of the CG6084 protein (blue in schematic, underlined in the amino acid sequence).
(TIF)

**S2 Fig. Generation of the *CG10638* mutant allele.** (**A**) CRISPR-mediated mutagenesis of the *CG10638* gene. A sgRNA was designed for the sequence within the exon common to the *CG10638* isoforms. The genomic map was adapted from FlyBase (http://flybase.org). (**B**) Breakpoint of the *CG10638*[10-1] allele. The *CG10638*[10-1] mutation caused a frameshift (yellow) leading to premature termination of all isoforms of the CG10638 protein. The mutant proteins lack most of the catalytic domain (blue in schematic, underlined in the amino acid sequence).
(TIF)

**S3 Fig. Generation of the *Sodh* mutant alleles.** (**A**) CRISPR-mediated mutagenesis of the *Sodh-1* gene. A sgRNA was designed for the sequence within the exon common to both isoforms of *Sodh-1*. The *Sodh-1*[14−1] mutation caused a frameshift (yellow) leading to premature termination of both isoforms of the Sodh-1 protein. The mutant proteins lack most of the catalytic domain (blue in schematic, underlined in the amino acid sequence). (**B**) CRISPR-mediated mutagenesis of the *Sodh-2* gene. A sgRNA was designed for the sequence in the third exon of *Sodh-2*. The *Sodh-2*[9−3] mutation caused a frameshift (yellow) leading to premature termination of the Sodh-2 protein. The mutant proteins lack most of the catalytic domain (blue in schematic, underlined in the amino acid sequence). The genomic maps in (A) and (B) were adapted from FlyBase (http://flybase.org).
(TIF)

**S4 Fig. Metabolic phenotypes of *AR* and *Sodh* mutants.** The amount of glucose, sorbitol, and fructose contained in the hemolymph of *AR* and *Sodh* mutant third instar larvae was measured using GC/MS. *OR* and *w* were used as controls; 10 larvae per batch, *n* = 5 batches for all experiments. Bar graphs show mean ± SE. The results of statistical significance tests between OR and *AR* or *Sodh* mutants are shown. $^*P < 0.05$; $^{**}P < 0.01$; $^{****}P < 0.0001$. The data underlying the graphs can be found in **S1 Data**. AR, aldose reductase; Sodh, sorbitol dehydrogenase. (TIF)

**S5 Fig. Phenotypes of *AR* and *Sodh* mutant larvae reared on regular diets.** (**A**, **B**) Larval body weight of *AR* and *Sodh* mutants; 10–50 animals per batch, *n* = 3 batches. Line graphs show mean ± SE. (**C**) Timing of pupal formation in *AR* and *Sodh* mutants. Thirty larvae were grown per vial, and the number of pupae was scored every 24 hours. *n* = 3 vials. The line graph shows mean ± SE. (**D**) Pupal volume of *AR* and *Sodh* mutants. *n* = 30 animals. Bars in the scatter plot indicate median. (**E**) Adult body weight of *AR* and *Sodh* mutants. Thirty animals per batch, *n* = 3 batches. The bar graph shows mean ± SE. The data underlying the graphs can be found in **S1 Data**. AR, aldose reductase; Sodh, sorbitol dehydrogenase. (TIF)

**S6 Fig. Rescue of *AR* and *Sodh* mutants by ubiquitous Mondo expression.** (**A–C**) Body weight, whole body triacylglyceride, and hemolymph glucose levels in larvae of control, *AR* mutant, *AR* mutant with ubiquitous *Mondo* overexpression (*AR, act > Mondo*). (**D–F**) Body weight, whole body triacylglyceride, and hemolymph glucose levels in larvae of control, *Sodh* mutant, and *Sodh* mutant with ubiquitous *Mondo* overexpression (*Sodh, act > Mondo*); 5–10 larvae per batch, *n* = 3 batches. Bar graphs show mean ± SE. $^*P < 0.05$; $^{**}P < 0.01$; $^{***}P < 0.001$; $^{****}P < 0.0001$. The data underlying the graphs can be found in **S1 Data**. AR, aldose reductase; Sodh, sorbitol dehydrogenase; TAG, triacylglyceride. (TIF)

**S7 Fig. Knockin of the Venus fluorescent protein in the *Mondo* locus.** (**A**) Schematic drawing of the Mondo locus (adapted from FlyBase, http://flybase.org). The Venus fluorescent protein was knocked-in at the C-terminus of the Mondo coding region (yellow). (**B**) Western blot using fat body extracts from the Mondo::Venus line. The Mondo::Venus fusion protein was detected with the anti-GFP polyclonal antibody. Original uncropped western blot image can be found in **S1 Raw** image. (TIF)

**S8 Fig. Analysis of ChREBP localization in mouse hepatocytes.** (**A**) CRISPR-mediated knockout of *Sord*. A crRNA was designed for the sequences in the intron 2 and the exon 9 of the *Sord* gene, resulting in the deletion from the exon 3 to the middle of the exon 9. (**B**) Frozen liver sections were stained with anti-CYP3A4 antibody to label pericentral hepatocytes. Regions of interest were set on the CYP3A4-positive area for quantification of ChREBP signals in pericentral hepatocytes. CV and PV are indicated in the picture. Scale bar represents 100 μm. ChREBP, carbohydrate response element-binding protein; CV, central vein; PV, portal vein. (TIF)

**S9 Fig. Transcriptomes of starved wild-type and *Sodh* mutant larvae.** A comparison of the transcriptomes of starved wild-type and *Sodh* mutant larvae; 30 larvae per batch, *n* = 3 batches. Correlation coefficient (*r*) is indicated in the plot. The data underlying the graphs can be found in **S9 Data**. (TIF)

**S1 Data. Data used to plot all graphs and to perform statistical analyses.** The Excel files contains the raw data to plot all graphs and the results of the statistical analysis. Each tab in the file shows the name of the figure panel produced based on the data displayed.
(XLSX)

**S2 Data. Sugar-responsive Mondo/Mlx-target genes.** Previous study has reported sugar-dependent transcriptomes in wild-type and mutants of *max-like protein X* (*mlx*, also known as *bigmax*), the obligated partner of Mondo [6]. To identify Mondo/Mlx-target genes, RNA-seq datasets reported in (GES70980) [6] were analyzed using the FlyBase reference genome (Dmel Release 6.19). First, we selected genes whose expression levels were significantly changed between control and *mlx* mutant larvae under high sugar conditions. Of those, genes whose expression levels were significantly different between control and *mlx* mutants under low sugar conditions were removed. control_HSD = average expression levels of triplicated experiments of HSD-fed control larvae (FPKM), mlx1_HSD = average expression levels of triplicated experiments of HSD-fed *mlx* mutant larvae (FPKM), logFC = log2 fold-change, test_stat = test statistics, p_value = uncorrected *p*-value of the test statistics, q_value = adjusted *p*-value of the test statistics with Benjamin–Hochberg correction.
(XLSX)

**S3 Data. Differential expression test data of Mondo/Mlx-target genes in starved and glucose-fed wild-type larvae.** WT_starved = average expression levels of triplicated experiments of starved wild-type larvae (FPKM), WT_glucose = average expression levels of triplicated experiments of glucose-fed wild-type larvae (FPKM), logFC = log2 fold-change, test_stat = test statistics, p_value = uncorrected *p*-value of the test statistics, q_value = adjusted *p*-value of the test statistics with Benjamin–Hochberg correction.
(XLSX)

**S4 Data. Differential expression test data of Mondo/Mlx-target genes in starved and sorbitol-fed wild-type larvae.** WT_starved = average expression levels of triplicated experiments of starved wild-type larvae (FPKM), WT_sorbitol = average expression levels of triplicated experiments of sorbitol-fed wild-type larvae (FPKM), logFC = log2 fold-change, test_stat = test statistics, p_value = uncorrected *p*-value of the test statistics, q_value = adjusted *p*-value of the test statistics with Benjamin–Hochberg correction.
(XLSX)

**S5 Data. Differential expression test data of Mondo/Mlx-target genes in starved and sorbitol-fed *Sodh* mutant larvae.** sodh_starved = average expression levels of triplicated experiments of starved *Sodh* mutant larvae (FPKM), sodh_sorbitol = average expression levels of triplicated experiments of sorbitol-fed *Sodh* mutant larvae (FPKM), logFC = log2 fold-change, test_stat = test statistics, p_value = uncorrected *p*-value of the test statistics, q_value = adjusted *p*-value of the test statistics with Benjamin–Hochberg correction.
(XLSX)

**S6 Data. Differential expression test data of Mondo/Mlx-target genes in starved and fructose-fed *Sodh* mutant larvae.** sodh_starved = average expression levels of triplicated experiments of starved *Sodh* mutant larvae (FPKM), sodh_fructose = average expression levels of triplicated experiments of fructose-fed *Sodh* mutant larvae (FPKM), logFC = log2 fold-change, test_stat = test statistics, p_value = uncorrected *p*-value of the test statistics, q_value = adjusted *p*-value of the test statistics with Benjamin–Hochberg correction.
(XLSX)

**S7 Data. Differential expression test data of metabolic genes in sugar-fed wild-type and *Sodh* mutant larvae.** Expression of metabolic genes was examined in starved and sugar-fed wild-type and *Sodh* mutant larvae. Log2 fold-changes between starved and sugar-fed larvae were calculated from average expression levels of triplicated experiments.
(XLSX)

**S8 Data. Differential expression test data of Mondo/Mlx-target genes in starved and glucose-fed *Sodh* mutant larvae.** sodh_starved = average expression levels of triplicated experiments of starved *Sodh* mutant larvae (FPKM), sodh_glucose = average expression levels of triplicated experiments of glucose-fed *Sodh* mutant larvae (FPKM), logFC = log2 fold-change, test_stat = test statistics, p_value = uncorrected *p*-value of the test statistics, q_value = adjusted *p*-value of the test statistics with Benjamin–Hochberg correction.
(XLSX)

**S9 Data. Gene expression data in starved wild-type and *Sodh* mutant larvae.** log10 FPKM (WT_starved) = average expression levels of triplicated experiments of starved wild-type larvae. log10 FPKM (sodh_starved) = average expression levels of triplicated experiments of starved *Sodh* mutant larvae. FPKM values were converted to log10 (FPKM).
(XLSX)

**S1 Raw image. Raw image of the western blot shown in S7B Fig.** The original uncropped western blot image of fat body extracts from the Mondo::Venus line.
(PDF)

## Acknowledgments

We thank Shu Kondo, the Bloomington Drosophila Stock Center, the Vienna Drosophila Resource Center, FlyORF in University of Zurich for supplying plasmids and fly stocks. We also thank Masayo Yamane, Miyuki Nishigata, Takumi Ichikawa, Shingo Usuki, Masatake Araki, Yoko Kimachi, Narumi Koga, and Yuki Takada for technical assistance, Naotada Ishihara and Takaya Ishihara for technical advice. We are also grateful to Daria Siekhaus, Prashanth Rangan, Takashi Koyama, and Yasushi Hiromi for critical reading of the manuscript.

## Author Contributions

**Conceptualization:** Hiroko Sano.

**Data curation:** Hiroko Sano.

**Formal analysis:** Hiroko Sano, Mariko Yamane, Hitoshi Niwa.

**Funding acquisition:** Hiroko Sano, Akira Nakamura, Masayasu Kojima.

**Investigation:** Hiroko Sano, Akira Nakamura, Takashi Nishimura, Kimi Araki, Kazumasa Takemoto, Kei-ichiro Ishiguro, Hiroki Aoki, Masayasu Kojima.

**Project administration:** Hiroko Sano.

**Resources:** Hiroko Sano, Akira Nakamura.

**Software:** Yuzuru Kato.

**Supervision:** Hiroko Sano, Akira Nakamura.

**Validation:** Hiroko Sano.

**Visualization:** Hiroko Sano.

**Writing – original draft:** Hiroko Sano.

**Writing – review & editing:** Hiroko Sano, Akira Nakamura, Mariko Yamane, Hitoshi Niwa, Takashi Nishimura, Kimi Araki, Kazumasa Takemoto, Kei-ichiro Ishiguro, Hiroki Aoki, Yuzuru Kato, Masayasu Kojima.

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
