## [Editor Report · Decision Letter 0]

25 Jan 2022

Dear Dr Sano, 

Thank you for submitting your Review Commons manuscript entitled "The polyol pathway is an evolutionarily conserved system for sensing glucose uptake" for consideration as a Research Article by PLOS Biology. Please accept my sincere apologies for the great delay in getting back to you as we consulted with an academic editor about your submission. 

Your manuscript has now been evaluated by the PLOS Biology editorial staff, as well as by an academic editor with relevant expertise, and I am writing to let you know that we would like to invite you to submit a revision that addresses the remaining reviewer comments. 

However, before we can invite a revision, we need you to complete your submission by providing the metadata that is required for full assessment. To this end, please login to Editorial Manager where you will find the paper in the 'Submissions Needing Revisions' folder on your homepage. Please click 'Revise Submission' from the Action Links and complete all additional questions in the submission questionnaire.

Once your full submission is complete, your paper will undergo a series of checks in preparation for peer review. Once your manuscript has passed the checks it will be sent out for review. To provide the metadata for your submission, please Login to Editorial Manager (https://www.editorialmanager.com/pbiology) within two working days, i.e. by Jan 27 2022 11:59PM.

Given the disruptions resulting from the ongoing COVID-19 pandemic, please expect some delays in the editorial process. We apologise in advance for any inconvenience caused and will do our best to minimize impact as far as possible.

Kind regards,

Richard

Richard Hodge, PhD

Associate Editor, PLOS Biology

rhodge@plos.org

PLOS

---

## [Editor Report · Decision Letter 1]

27 Jan 2022

Dear Dr Sano,

Thank you very much for submitting your manuscript "The polyol pathway is an evolutionarily conserved system for sensing glucose uptake" for consideration as a Research Article at PLOS Biology. As you know, your partly-revised manuscript and plan of revision have been evaluated by the PLOS Biology editors and by an Academic Editor with relevant expertise.

Based on your responses to the reviews from Reviews Commons, we would welcome re-submission of a revised version that takes into account the remaining comments from the reviewers. In addition, the Academic Editor has provided some additional comments regarding the in vivo mouse data (comments provided below my signature).

We cannot make any decision about publication until we have seen the revised manuscript and your response to the reviewers' comments. Your revised manuscript is also likely to be sent for further evaluation by the original Review Commons reviewers.

We expect to receive your revised manuscript within 3 months. Please email us (plosbiology@plos.org) if you have any questions or concerns, or would like to request an extension. At this stage, your manuscript remains formally under active consideration at our journal; please notify us by email if you do not intend to submit a revision so that we may end consideration of the manuscript at PLOS Biology.

**IMPORTANT - SUBMITTING YOUR REVISION**

*Re-submission Checklist*

*Published Peer Review*

*PLOS Data Policy*

*Blot and Gel Data Policy*

Sincerely,

Richard

Richard Hodge, PhD

Associate Editor, PLOS Biology

rhodge@plos.org

COMMENTS FROM THE ACADEMIC EDITOR:

The authors should show (1) body weights of the mice and (2) blood glucose and insulin levels at the time of tissue harvest. (3) Preferably, the authors should also show liver triglyceride levels.

---

## [Decision Letter · Decision Letter 2]

10 May 2022

Dear Dr Sano,

Thank you for submitting your revised Research Article entitled "The polyol pathway is an evolutionarily conserved system for sensing glucose uptake" for publication in PLOS Biology. I have now obtained advice from the original reviewers from Review Commons and have discussed their comments with the Academic Editor.

As you can see, the reviewers appreciated the additional data included in the revised manuscript to address their comments. Based on the reviews, I am pleased to say that we will probably accept this manuscript for publication, provided you address the remaining concerns outlined by Reviewer #2. In addition, please make sure to address the following data and other policy-related requests that I have provided below (points A-D):

(A) Please ensure that the data deposited in the GEO database (accession number GSE195759) is made publicly available at this stage.

(B) Please also ensure that each of the relevant figure legends in your manuscript include information on *WHERE THE UNDERLYING DATA CAN BE FOUND*, and ensure your supplemental data file/s has a legend.

(C) Thank you for already providing the raw image of the blot presented in Figure S7B. However, we note that this may not be the fully uncropped version. If this is the case, we ask that you please provide the original, uncropped and minimally adjusted image, even if the image contains data that is not relevant to this work. 

(D) Finally, please ensure that your Data Statement in the submission system accurately describes where your data can be found and is in final format, as it will be published as written there. This includes referring to underlying data provided in the Supplementary Information and the data deposited in the GEO. 

We expect to receive your revised manuscript within two weeks.

*Published Peer Review History*

*Early Version*

Sincerely,

Richard

Richard Hodge, PhD

Associate Editor, PLOS Biology

rhodge@plos.org

REVIEWS:

Reviewer #1: I am satisfied with the revision experiments performed.

Reviewer #2: Here, authors use data from experiments in Drosophila and mice to argue that the polyol pathway, which converts glucose to fructose via sorbitol, has a conserved function in glucose-sensing. Although this pathway is well conserved, it had not been previously attributed to glucose-sensing, and thus the results are potentially broadly important. 

The manuscript is a revised version, after being reviewed at Review Commons. The authors have made considerable improvements to the manuscript and have addressed my previous comments. I have only the following minor comments: 

1. When describing the phenotype on larval growth of the AR and Sodh mutants, the authors describe "weight loss" on several occasions, but actually it's just a decline in growth rate (there is not a loss or decline in the actual weight). 

2. The authors have mostly addressed my previous comment that the polyol pathway is not absolutely required (depends on fed/starved state) -- but there is still rather strong language in the abstract: lines 42-3 "depends on", 45-6 "is required". Consider softening/qualifying these statements. 

3. There is still something I don't understand in the discussion: you mention that the affinity of AR for glucose is very low compared to that of hexokinase (339-40), but then you say that "glucose flows into the polyol pathway before being metabolized to G6P..." (no reference for this statement by the way). Can you please explain how that works for me -- and/or maybe cite a reference? 

Reviewer #3 (Gabriel Leprivier, signs review): The authors adequately answered all my points.

---

## [Editor Report · Decision Letter 3]

17 May 2022

Dear Dr Sano,

On behalf of my colleagues and the Academic Editor, Rebecca Haeusler, I am pleased to say that we can accept your Research Article "The polyol pathway is an evolutionarily conserved system for sensing glucose uptake" for publication in PLOS Biology, provided you address any remaining formatting and reporting issues. These will be detailed in an email that will follow this letter and that you will usually receive within 2-3 business days, during which time no action is required from you. Please note that we will not be able to formally accept your manuscript and schedule it for publication until you have completed any requested changes.

PRESS

Sincerely,

Richard 

Richard Hodge, PhD

Associate Editor, PLOS Biology

rhodge@plos.org

PLOS
